# HCV Screening in a Sicilian Centre: A Descriptive Cohort Profile

**DOI:** 10.3390/v17091252

**Published:** 2025-09-16

**Authors:** Maria G. Minissale, Salvatore Petta, Fabio Cartabellotta

**Affiliations:** 1Internal Medicine Unit, Buccheri La Ferla Fatebenefratelli Hospital, Via Messina Marine 197, 90123 Palermo, Italy; fabiocartabellotta@gmail.com; 2Gastroenterology and Hepatology, Promise, Università di Palermo, 90127 Palermo, Italy; salvatore.petta@unipa.it

**Keywords:** screening, HCV

## Abstract

Introduction: Hepatitis C virus (HCV) infection prevalence in Italy varies according to geographical areas and clusters of infection. Moreover, epidemiological studies are old, and the actual prevalence of HCV active infections is also affected by the use of direct-acting antiviral therapies (DAAs) that achieve sustained virologic response (SVR) in >95% of treated patients. We aimed to evaluate the prevalence of HCV infections in in- or outpatients referred to a Sicilian hospital. Materials and methods: The study was conducted in the Buccheri La Ferla Hospital, in Palermo (Sicily), from 1 November 2019 to March 2022. We consecutively screened for HCV infections all inpatients who were evaluated on admission to the ward and all outpatients who referred to the central laboratory. All patients were screened using serological detection of HCV antibodies. Results: In the entire cohort, 469 out of 15,550 patients (3%) showed anti-HCV positivity, and this rate progressively increased according to classes of age (0.4% for <40 yrs, 3% for 40–60 yrs, 4% for >60–80 yrs, and 6.4% for >80 yrs). Among patients with anti-HCV positivity, 44.3% were HCV-RNA negative, 39.2% had HCV-RNA not available, and 16.4% were HCV-RNA positive. In total, 44.1% of patients with HCV-RNA positivity underwent DAA-based antiviral therapy. Conclusions: HCV screening programs can be useful in identifying infected patients at risk of liver disease progression and/or infection spreading. The implementation of laboratory strategies based on HCV reflex testing, the activation of dedicated linkage-to-care plans, and a focus on higher-risk groups could increase the effectiveness of screening programs.

## 1. Introduction

Hepatitis C virus (HCV) infection is a major public health problem worldwide, affecting more than 71 million people with a prevalence of roughly 1%, that varies greatly by region of the world. It is responsible for the development of cirrhosis and its complications such as liver-decompensation and hepatocellular carcinoma, and it is the cause of about 400,000 deaths per year [1].

Since 2014, the introduction of Direct-Acting Antivirals (DAAs) has revolutionized the management of HCV infection. With their short treatment duration (8–12 weeks), excellent safety profile—even in patients with decompensated liver disease—and sustained virological response rates exceeding 95%, DAAs have led to a marked reduction in both liver-related and extrahepatic complications [2,3,4].

According to this epidemiological and clinical burden and to the breakthrough related to DAAs, in May 2016, the World Health Organization (WHO) set ambitious targets to eliminate hepatitis C as a public health threat by 2030. Specifically, the key objectives included (1) the reduction in new infections by 90%, (2) the reduction in mortality by 65%, (3) the diagnosing of 90% of people living with HCV, and (4) the treatment of 80% of those diagnosed with HCV [5].

In this landscape, when looking at Italy, historically it has been considered one of the countries with the highest rate of HCV prevalence in Western Europe [6,7,8,9,10], and an old study in a small Sicilian town reported a 2002 prevalence of HCV infection of about 10% [11]. However, to date, the epidemiological scenario has probably changed, and a recent model, also considering DAA-treated patients, estimated, by January 2021, a prevalence of active HCV infection of 0.66% corresponding to about 398,610 infected individuals, with a prevalence of 0.72% in southern Italy. These data, however, highlight the concern related to the “undeclared” infected patients, which is the percentage of subjects who ignore their own positivity, and which is a source of contagion within the population. Along this line, Italy funded a free screening program for the micro-elimination of the HCV, identifying, in its first phase, as classes of subjects with a greater risk of infection, the cohort of those born between 1969 and 1989, subjects residing in prisons, and people with previous or active drug addiction who belong to Services for pathological Addictions [8]. The two cohorts (1969–1989) represent the dual epidemic: (1) an older cohort likely infected decades ago through transfusions or unsterile procedures (mainly in southern Italy), now presenting with cirrhosis and HCC and (2) a younger cohort, primarily PWID and MSM, accounting for most new infections.

Through awareness and screening programs, individuals at risk of a HCV infection should be directed toward curative treatment and educated on how to avoid key risk factors (such as needle sharing and high-risk sexual practices). This would help prevent the transmission of the virus to otherwise healthy individuals, making the intervention not only beneficial for the individual but also impactful at the community level.

All in all, we implemented a screening program, referred to the “Buccheri La Ferla” Hospital in Palermo. The primary aim of this study was to determine the prevalence of anti-HCV positivity and active viremia in a large cohort of inpatients and outpatients. We also aimed to identify the populations at highest risk for the micro-elimination of HCV.

## 2. Materials and Methods

This retrospective study was conducted in several sequential phases.

Phase 1: Information dissemination about the project was carried out within the hospital using materials such as posters and brochures.

Phase 2: From November 2019 to March 2022, all consecutive inpatients and outpatients were screened for HCV antibodies using immunoassay. Patients who tested positive were subsequently contacted and underwent quantitative HCV-RNA assessment using real time PCR (Xpert, Kit NeuMoDx HCV quantCE IVD, cut off 10–100.000.000 IU/mL).

Reflex testing was not used, but those who tested positive for anti-HCV had their HCV-RNA tested at a later time.

Inpatients included all patients admitted to the Medicine or Surgery wards, while outpatients comprised all consecutive patients referred to the hospital central laboratory.

Phase 3: Patients with active HCV infections (both anti-HCV and HCV-RNA positive) were called by a dedicated person (secretary), who informed them of the clinical situation and made an outpatient appointment with a dedicated hepatologist who examined and treated the patients.

Phase 4: Systematic collection of anamnestic, demographic, data was performed. These data were entered into a database for subsequent analysis.

This study received approval from the Hospital local Ethics Committee, and the informed consent was verbal.

Inclusion/exclusion criteria: included all patients > 18 years of age who were hospitalized, and all patients > 18 years of age who had external outpatient examinations and accepted the proposed sampling.

The variables collected were sex (male–female); age groups and average age in years; in- and outpatients; type of department involved (medicine, surgery, laboratory); anti-HCV (positive–negative); HCV-RNA (positive–negative).

SVR stands for sustained virologic response. It refers to the absence of detectable hepatitis C virus (HCV)-RNA in a patient’s blood 12 weeks or more after completing antiviral treatment. Achieving a SVR is considered a virologic cure for hepatitis C.

An active HCV infection occurs when the hepatitis C virus is present and replicating in the body. This is confirmed by the presence of HCV-RNA in the blood, indicating that the virus is currently circulating and may cause liver damage.

Spontaneous clearance refers to the natural elimination of the hepatitis C virus from the body without antiviral treatment. It occurs when the immune system successfully suppresses and removes the virus, confirmed by undetectable HCV-RNA tests.

## 3. Results

From November 2019 to March 2022 15,550 subjects of >18 years of age were evaluated. In total, 48.6% were males and the mean age was 55 years. The characteristics of patients stratified as inpatients or outpatients is reported in Table 1.

In the entire cohort, 15,081 patients (97%) had anti-HCV negativity, while 469 (3%) showed anti-HCV positivity. The prevalence of anti-HCV positivity was similar according to gender (3% in males and 2.9% in females), while it progressively increased according to classes of age (0.4% for <40 yrs, 3% for 40–60 yrs, 4% for >60–80 yrs, and 6.4% for >80 yrs) (Table 2). When splitting the population in outpatients and inpatients, the prevalence of anti-HCV positivity was 1.5% (103/6488) among the first, and 4% (366/9062) among the last. As for the entire cohort, and also in inpatients and outpatients considered separately, the prevalence of anti-HCV positivity was similar according to gender, while it progressively increased according to age (Table 2). Among inpatients, the prevalence of anti-HCV positivity was slightly higher in those admitted to Medicine compared with Surgery wards.

When focusing on the 469 patients with anti-HCV positivity, 208 out of them (44.3%) were HCV-RNA negative. This condition was attributable to sustained virological response (SVR) following antiviral therapy in 66 patients (31.7%), while the remaining 142 patients (68.3%) achieved spontaneous viral clearance (Table 3). HCV-RNA results were unavailable for 39.2% of patients, because of refusal to perform the test after the recall activity, and this proportion was higher among outpatients (65%) when compared with inpatients (32%), the percentage was similar between males and females; there was a high frequency of patients in the 60–80 age group (Table 3). Finally, 77 patients (16.4% of the anti-HCV + patients, and 0.4% of the entire cohort) were HCV-RNA positive, and this prevalence was higher among inpatients (18.5% of the anti-HCV + inpatients, and 0.7% of the entire inpatient cohort) with respect to outpatients (8.7% of the anti-HCV + outpatients, and 0.1% of the entire outpatient cohort) (Table 3). About half of all patients with HCV-RNA positivity (n = 34, 44.1%) underwent DAA-based antiviral therapy (Table 3). Specifically, 6 out of 9 (66.6%) outpatients underwent antiviral therapy with respect to 28 out of 68 (41.1%) inpatients (Table 3). Overall, treatment was not started because of (1) death in 10 patients; (2) old age, multiple advanced pathologies, decompensated cirrhosis, and/or advanced HCC in 23 patients; and (3) patient refusal in 10 patients.

## 4. Discussion

The availability of a safe and highly effective antiviral therapy for HCV infection led the WHO to set ambitious targets to eliminate hepatitis C as a public health threat by 2030. In certain regions of Southern Italy, a notably high prevalence of hepatitis C virus (HCV) positivity has been observed. This phenomenon is not limited to the Italian context but is also evident in other European countries. In nations such as Spain and Germany, epidemiological studies have highlighted significant rates of infection among socially vulnerable groups, including individuals experiencing homelessness, people who use drugs, and incarcerated populations. These findings suggest a strong correlation between social marginalization and the spread of the virus, underscoring the need for targeted interventions (micro-elimination) focused on prevention, early diagnosis, and equitable access to treatment [12,13,14,15].

Consistently, Italy funded a free screening program for HCV elimination, articulated in different phases according to age groups and clusters of individuals at higher risk. In this context, from November 2019 to March 2022, at the “Buccheri La Ferla” Sicilian hospital, we conducted a pilot HCV screening program and found that overall, 3% of screened individuals were anti-HCV positive. Notably, this rate was higher among inpatients (especially those admitted to medicine units) compared with outpatients and progressively increased with age. However, when looking at active HCV infection, only about 16% of anti-HCV positive individuals were HCV-RNA positive, corresponding to a prevalence of active infection of about 0.4%. We also observed that about half of anti-HCV positive patients had no active infection, 39% of anti-HCV positive patients refused HCV-RNA testing, and only 44% of patients with an active HCV infection underwent DAA-based antiviral therapy. A similar experience has been reported in another Italian center. An intra-hospital screening was conducted in Naples in 2020–2021, with a total of 12,665 screened patients [16]. Of these, 4% were anti-HCV positive, and among them, 23.1% had been previously treated, 33.9% were discharged before being tested for HCV-RNA, and 5.1% were not tested due to short life expectancy. When looking at the 38% of anti-HCV positive patients who were tested for HCV-RNA, about half (46.2%) were HCV-RNA positive and 95.6% of them started antiviral treatment.

The Italian-funded free screening for HCV elimination identified, in its first phase, the cohort of those born between 1969 and 1989 as the target group for screening. Our study reported a prevalence of anti-HCV positivity in this age cohort of about 3%, while the highest prevalence was observed in patients older than 60 years. Our data are similar to those from other Italian experiences. The Lombardy screening program in Northern Italy targeted subjects born between 1969 and 1989 using point-of-care (POC) testing, which was offered concomitantly with COVID-19 vaccination. Among the 7219 subjects born between 1969 and 1989 who underwent HCV screening through POC, 7 (0.10%) tested anti-HCV positive: 5 (0.07%) had confirmed anti-HCV positivity, and 4 of them (0.05%) were HCV-RNA positive by standard confirmation tests. Overall, these and our data raise some considerations about the age groups worthy of screening. In favor of initially screening younger individuals (born between 1969 and 1989), we can account for the cost-effectiveness of this strategy [8], the identification of young subjects with a long-life expectancy and a consequent higher risk of long-term complications, as well as the potential risk of spreading the infection across the community. However, on the other hand, focusing on the screening of older patients could allow the identification of a high proportion of HCV-infected patients who, even if sometimes with severe comorbidities and short life expectancy, remain at elevated risk of contributing to nosocomial transmission. Their frequent hospitalizations and exposure to invasive medical procedures underscore the importance of treatment not only for individual benefit, but also as a critical measure for infection control and public health protection.

Our study, as well as the aforementioned Italian studies, reported that less than half of anti-HCV positive patients had an active HCV infection. These data are consistent with Germany’s “Check-Up 35+” primary care screening program, where anti-HCV and HCV-RNA prevalence were 0.79% and 0.13%, respectively, showing that among patients with positive anti-HCV, only 16.3% were HCV-RNA positive [17].

These findings could be partially explained by the proportion of patients who underwent antiviral therapy, but they could also reflect the immune system’s ability to eradicate acute or chronic HCV infection. Further studies should investigate potential changes, compared with the past, in the immune system’s ability to control HCV infections.

In our study, as in the aforementioned Italian experience, we also reported that a significant proportion of patients with a positive anti-HCV test refused HCV-RNA testing, and some patients with active HCV infection did not undergo antiviral therapy.

In this context, in a real-world polish population of nearly 19,000 HCV-infected patients, a 2.7% loss to follow-up rate was documented. Independent predictors of this phenomenon were male gender, GT3 infection, HIV co-infection, alcohol addiction, mental illnesses, lack of prior antiviral treatment, and discontinuation of DAA therapy [18].

A nationwide micro-elimination strategy accurately mapped the ever-diagnosed HCV population in the Netherlands and indicated that 27% of LTFU HCV-infected patients re-linked to care had advanced fibrosis or cirrhosis, emphasizing the potential value of systematic retrieval for HCV elimination [19].

These data highlight that for a screening program to be effective, it needs to be implemented alongside strategies that simplify the patient pathway. Specifically, the introduction of reflex testing (automatic HCV-RNA testing in the blood sample of anti-HCV positive patients) and active, simplified programs of linkage-to-care that facilitate access for HCV-RNA positive patients to centers for HCV treatment could make screening programs more effective and useful.

The linkage-to-care model, like that we have in Sicily, implemented an integrated approach that connects laboratory diagnostics with hospital-based clinical evaluation. Within this system, biological samples are collected in laboratory settings and, in the event of a positive result for HCV infection, patients are promptly notified—typically via SMS or email. They are then referred to specialized hospital clinics dedicated to the management and treatment of hepatitis C. This streamlined pathway facilitates timely access to care and enhances the continuity between diagnosis and therapeutic intervention.

Our study has several limitations. Specifically, it is a singe center study that limited generalizability, lacked risk factor data, had no liver disease staging, and had a high proportion of missing RNA results, which introduces possible selection bias.

The strengths are a large sample size, a real-world setting, and the inclusion of both in- and outpatients.

## 5. Conclusions

This study highlights the high prevalence of HCV-positive individuals within hospital settings (around 3%), underscoring the importance of implementing hospital-based screening. Such screening should not be an “una tantum” initiative, but rather a continuous assessment among hospitalized patients, who are often older and more vulnerable. These individuals face a higher risk of infection and may also contribute to viral transmission, making ongoing screening a crucial strategy for effective prevention and control.

In conclusion, our pilot study in a Sicilian hospital—spanning both inpatient and outpatient populations—demonstrates the untapped potential of hospital-based HCV screening as a strategic entry point for identifying undiagnosed infections and initiating timely care. By embedding reflex testing protocols into routine diagnostics and activating dedicated linkage-to-care pathways, hospitals can serve not only as clinical settings but as pivotal hubs for public health intervention. However, the success of such programs hinges on our ability to confront persistent barriers across the care continuum: fragmented follow-up systems, limited patient engagement, and gaps in specialist access continue to undermine the impact of early diagnosis. To truly curb disease progression and transmission, screening must be paired with structural reforms that ensure continuity of care—especially for high-risk and marginalized groups. This study adds to growing evidence that with the right infrastructure, hospital-based screening can evolve from a passive detection tool into a proactive engine for HCV elimination.

## Figures and Tables

**Table 1 viruses-17-01252-t001:** Characteristics of the cohort, overall and split by Outpatients and Inpatients.

	Entire Cohort N = 15,550	Outpatients N = 6488	Inpatients N = 9062
Males	7567 (48.6%)	2593 (40%)	4088 (54.9%)
Females	7983 (51.4%)	3895 (60%)	4974 (45.1%)
<40 years	5035 (32.4%)	4281 (66%)	754 (8.3%)
40–60 years	3084 (19.8%)	1218 (18.7%)	1866 (20.6%)
60–80 years	5323 (34.2%)	869 (13.3%)	4454 (49.1%)
>80 years	2108 (13.6%)	120 (2%)	1988 (22%)
Mean Age	55	38	67
Medicine Surgery Laboratory	5632 (36.2%) 3430 (22.1%) 6488 (41.7%)	6488	5632 (62.1%) 3430 (37.9%)

**Table 2 viruses-17-01252-t002:** Prevalence of anti-HCV positivity overall and in subgroups of patients.

	Entire Cohort N = 15,550	Outpatients N = 6488	Inpatients N = 9062
Prevalence of anti-HCV +	469 (3%)	103 (1.5%)	366 (4%)
Prevalence of anti-HCV +			
Males	232/7567 (3%)	51/2593 (1.9%)	181/4974(3.6%)
Females	237/7983 (2.9%)	52/3895 (1.3%)	185/4088 (4.5%)
Prevalence of anti-HCV +			
<40 years	22/5035 (0.4%)	17/4281(0.39%)	5/754 (0.6%)
40–60 years	93/3084 (3%)	40/1218 (3.2%)	53/1866 (2.8%)
60–80 years	217/5323 (4%)	36/869 (4%)	181/4454 (4%)
>80 years	137/2108 (6.4%)	10/120 (8%)	127/1988 (6.3%)
Prevalence of anti-HCV + Medicine Surgery			245/5632 (4.3%) 121/3430 (3.5%)

**Table 3 viruses-17-01252-t003:** Prevalence of active HCV infection among subgroups of patients with anti-HCV positivity.

	Entire cohort Anti-HCV + N = 469	Outpatients Anti-HCV + N = 103	Inpatients Anti-HCV + N = 366
**HCV-RNA** quantitative Positive DAA Negative Not available	77 (16.4%) 34 treated 208 (44.3%) 184 (39.2%)	9 (8.7%) 6 treated 1 not treatable 2 refused 27 (26.3%) 10 SVR 67 (65%)	68 (18.5%) 28 treated 10 died 22 not treatable 8 refused 181 (49.5%) 56 SVR 117 (32%)

## Data Availability

Data are contained within the article.

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
