# Peer review of "HCV Screening in a Sicilian Centre: A Descriptive Cohort Profile"

_viruses, 2025, doi:10.3390/v17091252_

Round 1
Reviewer 1 Report
Comments and Suggestions for Authors
In this paper, the Authors described a single center experience concerning the screening of HCV infection based on serum antibody detection, in a population of inpatients admitted to a Sicilian Hospital and outpatients who referred to the central laboratory of the same Hospital.
Three% of the entire cohort resulted anti-HCV positive and 0.4% were viremic; 44% of HCV RNA positive patients were treated, whereas the majority of HCV RNA positive patients didn’t receive any antiviral therapy due to death or multiple liver- and non-liver-related advanced pathologies, and patient’s refusal in a small group.
Results are in line with literature data, confirming opportunities and limitations of ongoing screening programs. Study limitations have been defined.
A careful reading of the text and tables is suggested (see page 2 “objectives include…” and table 1 “età media”.
Author Response
Comments 1 A careful reading of the text and tables is suggested (see page 2 “objectives include…” and table 1 “età media”).
Response 1 It can be added at the end of the introduction ”The primary aim of this study was to determine the prevalence of anti-HCV positivity and active viremia in a large cohort of in-patients and outpatients. We also aimed to identify the populations at highest risk”.
Eta media can be traslated in Median Age
Reviewer 2 Report
Comments and Suggestions for Authors
Major:
Abstract
- The sentence "39.2% had HCV-RNA not available..." does not explain the underlying reasons, which are crucial to interpreting the result. Was it due to patient refusal, loss to follow-up, or logistical issues? This must be clarified.
- The conclusion is vague: "HCV screening programs can be useful..." does not reflect the strength of the data. A stronger, more accurate version could be: "While hospital-based screening identifies a notable reservoir of anti-HCV positive individuals, our findings reveal critical bottlenecks in the cascade of care, with a large proportion of patients lost before confirmation of active infection and a low treatment uptake among those diagnosed.”
Introduction
- The final paragraph does not clearly state the study’s objectives. Instead of: “All in all, we implemented a screening program…”, a clearer aim could be: “The primary aim of this study was to determine the prevalence of anti-HCV positivity and active viremia in a large cohort of in-patients and outpatients. We also aimed to identify the populations at highest risk and key barriers in the local cascade of care, from diagnosis to treatment”
- The introduction should better emphasize the shift from a therapeutic to a public health challenge: the main question is no longer how to cure HCV, but who to treat and how to link them to care.
- The rationale behind the Italian risk-based screening criteria (birth cohort 1969–1989, PWID) is not discussed. It would be valuable to briefly explain the dual epidemic (1), an older cohort likely infected decades ago through transfusions or unsterile procedures (mainly in southern Italy), now presenting with cirrhosis and HCC; (2) a younger cohort, primarily PWID and MSM, accounting for most new infections. This epidemiological background is important for interpreting the study’s age-based findings.
- The concept of micro-elimination should be introduced, as this study is an example of HCV elimination efforts in a defined setting.
- The introduction assumes hospital-based screening is effective, but it should raise the question: How efficient is this strategy in practice? Hospitals gather patients with higher morbidity and frequent healthcare interactions, but actual prevalence and linkage-to-care outcomes are often unknown. This gap should be addressed.
Material and Methods. This section needs to be completely rewritten to provide essential methodological details.
- Study design: Was this retrospective or prospective? It is unclear.
- Inclusion/exclusion criteria: Only “>18 years” is mentioned. Were patients already known to be HCV-positive excluded? What about those who declined participation? This must be clarified.
- Informed consent: Was it written, verbal, or waived as part of standard care with opt-out? This is ethically important. Include the full name of the ethics committee and approval number
- Lab methods: Merely stating "immunoassay" and "PCR" is insufficient. Specify manufacturer, kit name, model, and cut-off values
- Reflex testing: Was this used? The text suggests not, which could explain the high rate of missing HCV-RNA results. This is a methodological issue that must be addressed
- Call center linkage: Vague description. Who called the patients? What information was provided? How were appointments scheduled? What clinical evaluation was performed?
- Definitions and data: Which variables were collected? Define key terms (e.g., SVR, active infection). Describe how spontaneous clearance was determined
- Statistical analysis: Completely missing. This must be addressed.
Results
- Table 1: No statistical comparisons between in-patients and outpatients. These groups are clearly different (e.g., age 67 vs. 38). Statistical testing is needed.
- Table 2: Prevalence of anti-HCV is 4% (in-patients) vs. 1.5% (outpatients). This is a key message. Is the difference statistically significant? Same for age trends: is the increase from 0.4% (<40 yr) to 6.4% (>80yr) significant?
- Table 3: Who are the patients with “HCV-RNA not available”? Compare their characteristics to those with RNA results. Likely more common among outpatients due to loss to follow-up. Moreover, what factors are associated with active infection among anti-HCV positives? Logistic regression would help. Among viremic patients, who remained untreated? Were they older, with more comorbidities? Quantify this.
Conclusions. This section should be split into “Discussion” and “Conclusions,” and follow a clearer structure: 1-Main findings; 2-Comparison with existing literature; 3-Interpretation and implications; 4-Strengths and limitations; and 5-Conclusions
- Compare with other cohorts. Southern Italy has a higher prevalence due to historical reasons. Compare viremia rates with studies from different European countries (e.g., Germany, Spain). Consider citing: PMID: 39453657; 39027941.
- The high prevalence in older patients is framed as a success. But since most untreated viremic patients were elderly with comorbidities, this raises an important question: is hospital screening identifying mostly frail individuals for whom treatment offers limited benefit? This nuance should be addressed. Treating them may still be important from a public health perspective, due to the high risk of nosocomial transmission.
- The 39.2% missing RNA results are the most striking finding. The discussion should go beyond recommending reflex testing. Who are the patients lost to follow-up? What do other studies say about barriers like chaotic lifestyles, low health literacy, or fear of diagnosis?
- The recommendation of “dedicated linkage-to-care plans” is too vague. Cite models shown to be effective in similar populations, e.g., patient navigators, integrated addiction/hepatology clinics, telemedicine, and mobile units.
- Strengths: Large sample size, real-world setting, inclusion of both in- and outpatients. Address
- Limitations: Single-center, limited generalizability, lack of risk factor data, no liver disease staging, and a high proportion of missing RNA results, which introduces possible selection bias. Address
- The conclusion should be more impactful. Consider expanding on the abstract suggestion, highlighting both the potential of hospital-based screening and the urgent need to address major barriers in the care cascade.
Minor
- In “use of direct action antiviral therapies (DAAs)”, the correct terminology is “direct-acting antiviral therapies (DAAs)”.
- In “all outpatients who affeered to the…”, “affeered” is incorrect. It should be “referred”.
- “admitted to medicine respect too surgery wards” is grammatically incorrect. It should be corrected to “compared to” or “relative to”.
- “39.2% had HCVRNA not available” contains two issues: the proper format is “HCV-RNA”, and a clearer phrasing would be: “HCV-RNA results were unavailable for 39.2% of patients”
- “it is cause of about 400,000 deaths per year” should be revised to: “it is the cause of about 400,000 deaths per year” or “it causes about 400,000 deaths per year”
- “3) the diagnosing of 90% of people living with HCV, and 3) the treatment of 80% of those diagnosed with HCV”, the numbering is incorrect. Please revise accordingly.
- “this condition being related to SVR after antiviral therapy in 66 patients (31.7%) and to spontaneous virological clearance in the remaining 142 patients (68.3%)” is difficult to follow. Consider rephrasing.
- “Since 2014, the availability of Direct Antiviral Agents (DAAs)...” is too long and would benefit from being split into two sentences for clarity.
- The Italian term “Età media” should be translated into English for consistency (e.g., “Mean age”).
- References: There are inconsistencies in the formatting. For example, “et all” should be corrected to “et al.” A thorough review of reference formatting is needed to ensure uniformity. The reference list appears limited. The authors should consider including more recent and comparative studies to strengthen the background and discussion.
Author Response
Abstract
- Comments 1 The sentence "39.2% had HCV-RNA not available..." does not explain the underlying reasons, which are crucial to interpreting the result. Was it due to patient refusal, loss to follow-up, or logistical issues? This must be clarified
Response 1: It was described that 39.2 %f patients had HCV-RNA not available because of refusing to perform the test after the recall activity, some patients were lost to follow up, others died
- Comments 2 The conclusion is vague: "HCV screening programs can be useful..." does not reflect the strength of the data. A stronger, more accurate version could be: "While hospital-based screening identifies a notable reservoir of anti-HCV positive individuals, our findings reveal critical bottlenecks in the cascade of care, with a large proportion of patients lost before confirmation of active infection and a low treatment uptake among those diagnosed.”
Response 2: The conclusions of the abstract are not vague, but concise and synthetic
Introduction
- Comments 3 The final paragraph does not clearly state the study’s objectives. Instead of: “All in all, we implemented a screening program…”, a clearer aim could be: “The primary aim of this study was to determine the prevalence of anti-HCV positivity and active viremia in a large cohort of in-patients and outpatients. We also aimed to identify the populations at highest risk and key barriers in the local cascade of care, from diagnosis to treatment”
Response 3: I agree.
It can be changed in ..”The primary aim of this study was to determine the prevalence of anti-HCV positivity and active viremia in a large cohort of in-patients and outpatients. We also aimed to identify the populations at highest risk”.
- Comments 4 The introduction should better emphasize the shift from a therapeutic to a public health challenge: the main question is no longer how to cure HCV, but who to treat and how to link them to care.
Response 4 Through national screening programs, individuals who are positive for HCV should be identified, especially those who are more fragile and at high risk. These individuals are then started on a therapy program.
- Comments 5
The rationale behind the Italian risk-based screening criteria (birth cohort 1969–1989, PWID) is not discussed. It would be valuable to briefly explain the dual epidemic (1), an older cohort likely infected decades ago through transfusions or unsterile procedures (mainly in southern Italy), now presenting with cirrhosis and HCC; (2) a younger cohort, primarily PWID and MSM, accounting for most new infections. This epidemiological background is important for interpreting the study’s age-based findings.
Response 5 I agree.
We can add that the two courts represent: (1), an older cohort likely infected decades ago through transfusions or unsterile procedures (mainly in southern Italy), now presenting with cirrhosis and HCC; (2) a younger cohort, primarily PWID and MSM, accounting for most new infections.
- Comments 6 The concept of micro-elimination should be introduced, as this study is an example of HCV elimination efforts in a defined setting.
Response 6 Microelimination is a strategy that aims to eliminate the virus from specific segments of the population, rather than an entire nation, such as drug addicts or, in our case, hospitalized patients.
- Comments 7 The introduction assumes hospital-based screening is effective, but it should raise the question: How efficient is this strategy in practice? Hospitals gather patients with higher morbidity and frequent healthcare interactions, but actual prevalence and linkage-to-care outcomes are often unknown. This gap should be addressed.
Response 7: Even with comorbidities, antiviral therapy is well tolerated, although some patients refuse treatment.
Material and Methods. This section needs to be completely rewritten to provide essential methodological details.
- Comments 8 Study design: Was this retrospective or prospective? It is unclear.
Response8: prospective
- Comments 9 Inclusion/exclusion criteria: Only “>18 years” is mentioned. Were patients already known to be HCV-positive excluded? What about those who declined participation? This must be clarified.
Response 9 Inclusion/exclusion criteria: all patients > 18 years of age who were hospitalized, and all patients > 18 years of age who had external outpatient examinations and accepted the proposed sampling.
- Comments 10 Informed consent: Was it written, verbal, or waived as part of standard care with opt-out? This is ethically important. Include the full name of the ethics committee and approval number
Response 10: the informed consent is verbal . The hospital local ethics commitee
- Comments 11 Lab methods: Merely stating "immunoassay" and "PCR" is insufficient. Specify manufacturer, kit name, model, and cut-off values
Response 11 PCR real time Xpert, Kit NeuMoDx HCV quantCE IVD , cut off 10- 100.000.000 IU/ml
- Comments 12 Reflex testing: Was this used? The text suggests not, which could explain the high rate of missing HCV-RNA results. This is a methodological issue that must be addressed
Response 12 Reflex testing was not used, but those who tested positive for anti-HCV had their HCV RNA tested at a later time
- Comments 13 Call center linkage: Vague description. Who called the patients? What information was provided? How were appointments scheduled? What clinical evaluation was performed?
Response 13 Patients were called by a dedicated person (secretary), who informed them of the clinical situation and made an outpatient appointment with a dedicated hepatologist who examined and treated the patients.
- Comments 14 Definitions and data: Which variables were collected? Define key terms (e.g., SVR, active infection). Describe how spontaneous clearance was determined
Response 14
The variables collected were: sex (male-female); age groups and average age in years ,in and out patients; type of department involved (medicine, surgery, laboratory); Anti- HCV (positive-negative); HCV-RNA (positive-negative).
SVR stands for Sustained Virologic Response. It refers to the absence of detectable hepatitis C virus (HCV) RNA in a patient’s blood 12 weeks or more after completing antiviral treatment. Achieving SVR is considered a virologic cure of hepatitis C
Active HCV infection occurs when the hepatitis C virus is present and replicating in the body. This is confirmed by the presence of HCV RNA in the blood, indicating that the virus is currently circulating and may cause liver damage
Spontaneous clearance refers to the natural elimination of the hepatitis C virus from the body without antiviral treatment. It occurs when the immune system successfully suppresses and removes the virus, confirmed by undetectable HCV RNA tests
- Comments 15 Statistical analysis: Completely missing. This must be addressed.
Response 15 The text explains the data in the tables in a descriptive manner without statistical evaluation.
Results
- Comments 16-17
Table 1: No statistical comparisons between in-patients and outpatients. These groups are clearly different (e.g., age 67 vs. 38). Statistical testing is needed.
Table 2: Prevalence of anti-HCV is 4% (in-patients) vs. 1.5% (outpatients). This is a key message. Is the difference statistically significant? Same for age trends: is the increase from 0.4% (<40 yr) to 6.4% (>80yr) significant?
Response 16-17
The text explains the data in a descriptive manner without statistical evaluation
- Comments 18
Table 3: Who are the patients with “HCV-RNA not available”? Compare their characteristics to those with RNA results. Likely more common among outpatients due to loss to follow-up. Moreover, what factors are associated with active infection among anti-HCV positives? Logistic regression would help.
Among viremic patients, who remained untreated? Were they older, with more comorbidities? Quantify this.
Response 18
The characteristics of patients with unavailable HCV RNA were:
HCV RNA was unavailable in 65% of outpatients and 32% of inpatients; the percentage was similar between males and females; there was a high frequency of patients in the 60-80 age group.
Viremic patients who remained untreated were older, with more severe comorbidities.
22 of them were not treatable, while 10 died and 8 refused treatment.
Conclusions.
This section should be split into “Discussion” and “Conclusions,” and follow a clearer structure:
1-Main findings; 2-Comparison with existing literature; 3-Interpretation and implications; 4-Strengths and limitations; and 5-Conclusions
- Comments 19
Compare with other cohorts. Southern Italy has a higher prevalence due to historical reasons. Compare viremia rates with studies from different European countries (e.g., Germany, Spain). Consider citing: PMID: 39453657; 39027941.
Response 19
In Southern Italy there is a high prevalence of HCV positivity, this also occurs in other European countries, such as Spain and Germany, where a high prevalence has been observed among the homeless, among drug users and in prisoners.
Biblio :
- Ryan, J. Valencia , G.Cuevas et al. Decrease in active hepatitis C infection among people who use drugs in Madrid, Spain, 2017 to 2023: a retrospective study. Euro Surveill;29(29):2300712.
- Ryan , J. Valencia , D. Sepúlveda-Crespo et al. Prevalence of HCV Infection Among People Experiencing Homelessness in Madrid, Spain. JAMA Netw 2024;7(10):e2438657.
- Sperle, G, Steffen , S.A. Leendertz et al. . Prevalence of Hepatitis B, C, and D in Germany: Results From a Scoping Review .Front Public Health2020 ; 28:8:424.
- Stöver, A Dichtl , D Schäffer et al. . HIV and HCV among drug users and people living in prisons in Germany 2022: WHO elimination targets as reflected in practice Harm Reduct J2023 Apr 13;20(1):5
- Comments 20
The high prevalence in older patients is framed as a success. But since most untreated viremic patients were elderly with comorbidities, this raises an important question: is hospital screening identifying mostly frail individuals for whom treatment offers limited benefit? This nuance should be addressed. Treating them may still be important from a public health perspective, due to the high risk of nosocomial transmission.
Response 20
Hospitalized patients are often elderly with comorbidities, however treating them means avoiding the risk of virus transmission, as they are subjects who still need healthcare (who undergo blood tests, intravenous therapies, etc.).
- Comments 21
The 39.2% missing RNA results are the most striking finding. The discussion should go beyond recommending reflex testing. Who are the patients lost to follow-up? What do other studies say about barriers like chaotic lifestyles, low health literacy, or fear of diagnosis?
Response 21
In a real-world polish population of nearly 19,000 HCV-infected patients, it was documented a 2.7% loss to follow-up rate. Independent predictors of this phenomenon were male gender, GT3 infection, HIV co-infection, alcohol addiction, mental illnesses, lack of prior antiviral treatment and discontinuation of DAA therapy.
D.Z.Michaluk , M.Brzdęk , O. Tronina . et al. Loss to follow-up of patients after antiviral treatment as an additional barrier to HCV elimination. BMC Med 2024;22:486.
A nationwide micro-elimination strategy accurately mapped the ever-diagnosed HCV population in the Netherlands and indicates that 27% of LTFU HCV-infected patients re-linked to care have advanced fibrosis or cirrhosis, emphasizing the potential value of systematic retrieval for HCV elimination.
Cas J Isfordink , M. van Dijk , S M Brakenhoff . Hepatitis C Elimination in the Netherlands (CELINE): How nationwide retrieval of lost to follow-up hepatitis C patients contributes to micro-elimination. Eur J Intern Med2022 Jul:101:93-97.
- Comments 22
The recommendation of “dedicated linkage-to-care plans” is too vague. Cite models shown to be effective in similar populations, e.g., patient navigators, integrated addiction/hepatology clinics, telemedicine, and mobile units.
Response 22
Linkage-to-care models (like the one we have in Silicia) could be connected to integrated laboratory-hospital evaluation systems, in which samples are taken in the laboratory, and in case of positive results for HCV infection, patients are contacted ( with SMS or mail) and sent to hospitals with clinics dedicated to the treatment of hepatitis.
- Comments 23
Strengths: Large sample size, real-world setting, inclusion of both in- and outpatients. Address
Response 23. We can add these strengths
- Comments 24
Limitations: Single-center, limited generalizability, lack of risk factor data, no liver disease staging, and a high proportion of missing RNA results, which introduces possible selection bias. Address
Response 24 We can add these limitations
- Comments 25
The conclusion should be more impactful. Consider expanding on the abstract suggestion, highlighting both the potential of hospital-based screening and the urgent need to address major barriers in the care cascade.
Response 25
In conclusion, our pilot study in a Sicilian hospital—spanning both inpatient and outpatient populations—demonstrates the untapped potential of hospital-based HCV screening as a strategic entry point for identifying undiagnosed infections and initiating timely care. By embedding reflex testing protocols into routine diagnostics and activating dedicated linkage-to-care pathways, hospitals can serve not only as clinical settings but as pivotal hubs for public health intervention.
However, the success of such programs hinges on our ability to confront persistent barriers across the care continuum: fragmented follow-up systems, limited patient engagement, and gaps in specialist access continue to undermine the impact of early diagnosis. To truly curb disease progression and transmission, screening must be paired with structural reforms that ensure continuity of care—especially for high-risk and marginalized groups.
This study adds to growing evidence that with the right infrastructure, hospital-based screening can evolve from a passive detection tool into a proactive engine for HCV elimination.
Minor
- Comments 26 In “use of direct action antiviral therapies (DAAs)”, the correct terminology is “direct-acting antiviral therapies (DAAs)”.
Response 26 I agree
- Comments 27
In “all outpatients who affeered to the…”, “affeered” is incorrect. It should be “referred”.
Response 27 I agree
- Comments 28
“admitted to medicine respect too surgery wards” is grammatically incorrect. It should be corrected to “compared to” or “relative to”.
Response 28 I agree with compared to
- Comments 29
“39.2% had HCVRNA not available” contains two issues: the proper format is “HCV-RNA”, and a clearer phrasing would be: “HCV-RNA results were unavailable for 39.2% of patients”
Response 29 I agree with HCV-RNA results were unavailable for 39.2% of patients
- Comments 30
“it is cause of about 400,000 deaths per year” should be revised to: “it is the cause of about 400,000 deaths per year” or “it causes about 400,000 deaths per year”
Response 30 I agree with “it is the cause of about 400,000 deaths per year”
- Comments 31
“3) the diagnosing of 90% of people living with HCV, and 3) the treatment of 80% of those diagnosed with HCV”, the numbering is incorrect. Please revise accordingly.
Response 31
It should be corrected in “3) the diagnosing of 90% of people living with HCV, and 4) the treatment of 80% of those diagnosed with HCV”,
- Comments 32
“this condition being related to SVR after antiviral therapy in 66 patients (31.7%) and to spontaneous virological clearance in the remaining 142 patients (68.3%)” is difficult to follow. Consider rephrasing.
Response 32
This condition was attributable to sustained virological response (SVR) following antiviral therapy in 66 patients (31.7%), while the remaining 142 patients (68.3%) achieved spontaneous viral clearance.
- Comments 33
“Since 2014, the availability of Direct Antiviral Agents (DAAs)...” is too long and would benefit from being split into two sentences for clarity.
Response 33
Since 2014, the introduction of Direct-Acting Antivirals (DAAs) has revolutionized the management of HCV infection. With their short treatment duration (8–12 weeks), excellent safety profile—even in patients with decompensated liver disease—and sustained virological response rates exceeding 95%, DAAs have led to a marked reduction in both liver-related and extrahepatic complications.
- Comments 34
The Italian term “Età media” should be translated into English for consistency (e.g., “Mean age”)
Response 34 I agree
- Comments 35
References: There are inconsistencies in the formatting. For example, “et all” should be corrected to “et al.” A thorough review of reference formatting is needed to ensure uniformity. The reference list appears limited. The authors should consider including more recent and comparative studies to strengthen the background and discussion.
Response 35 I agree , it can be changed with “et al.”
You can add these references
- Ryan , J. Valencia , G.Cuevas et al. Decrease in active hepatitis C infection among people who use drugs in Madrid, Spain, 2017 to 2023: a retrospective study. Euro Surveill.
- ;29:2300712.
- Ryan , J. Valencia , D. Sepúlveda-Crespo et al. Prevalence of HCV Infection Among People Experiencing Homelessness in Madrid, Spain. JAMA Netw. 2024; e2438657.
- Sperle, G, Steffen , S.A. Leendertz et al. . Prevalence of Hepatitis B, C, and D in Germany: Results From a Scoping Review. Front Public Health. 2020; 28:8:424.
- Stöver, A Dichtl , D Schäffer et al. . HIV and HCV among drug users and people living in prisons in Germany 2022: WHO elimination targets as reflected in practice Harm Reduct J. 2023;20:50.
Round 2
Reviewer 2 Report
Comments and Suggestions for Authors
Thank you for your thorough revisions and for addressing many of the points raised in my initial review. The manuscript has clearly improved, particularly in the level of methodological detail and in the overall structure of the discussion.
That said, there remain several key areas that require further attention before the work can meet the standards for publication. Based on my assessment, I am recommending that the manuscript undergo additional revision. The final decision on how to proceed will, of course, rest with the Editor.
Thank you again for your efforts and for your contribution to this important topic.
Author Response
- Comment 1 A major methodological flaw remains: the authors have declined to perform any statistical analysis, stating that their work is purely descriptive.In its current form, the manuscript constitutes a descriptive report rather than an analytical study. If the journal's scope accommodates such descriptive contributions, the title should be amended accordingly to ensure transparency for the reader. Possible alternatives might include: "A descriptive report on an HCV screening program in a Sicilian hospital" or "HCV screening in a Sicilian centre: A descriptive cohort profile" or similar. If, on the other hand, the editor considers statistical analysis necessary, I note that the current manuscript presents observed differences (such as the higher prevalence in inpatients versus outpatients or the trend with increasing age) as conclusive findings. Without statistical testing, these remain anecdotal observations and may be attributable to random variation rather than genuine effects.
Response 1 : I agree with changing in “HCV screening in a Sicilian centre: A descriptive cohort profile”.
- Comment 4: Emphasizing in the introduction the shift in HCV management from a therapeutic to a public health challenge
Response 4: In the introduction, HCV elimination is presented as a public health challenge and a key objective of the WHO. It outlines the types of patients to be treated, the two main cohorts, and individuals at risk. It could be further emphasized that...Through awareness and screening programs, individuals at risk of HCV infection should be directed toward curative treatment and educated on how to avoid key risk factors (such as needle sharing and high-risk sexual practices). This would help prevent the transmission of the virus to otherwise healthy individuals, making the intervention not only beneficial for the individual but also impactful at the community level.
- Comment 7: Framing the study as an evaluation of the efficiency of hospital-based screening.
Response 7 This study highlights the high prevalence of HCV-positive individuals within hospital settings (around 3%), underscoring the importance of implementing hospital-based screening. Such screening should not be a “una tantum” initiative, but rather a continuous assessment among hospitalized patients, who are often older and more vulnerable. These individuals face a higher risk of infection and may also contribute to viral transmission, making ongoing screening a crucial strategy for effective prevention and control.
- Comment 20: Discussing the clinical paradox of identifying mostly frail, elderly individuals, for whom treatment decisions are particularly complex.
Response 20 focusing on the screening of older patients could allow the identification of a high proportion of HCV-infected patients who, even if sometimes with severe comorbidities and short life expectancy, remain at elevated risk of contributing to nosocomial transmission. Their frequent hospitalizations and exposure to invasive medical procedures underscore the importance of treatment not only for individual benefit, but also as a critical measure for infection control and public health protection.

Round 3
Reviewer 2 Report
Comments and Suggestions for Authors
Thank you for your constructive revisions. The changes made have successfully addressed the key issues raised, and the manuscript is now improved. I have no further comments.